# Core-Shell Magnetic Imprinted Polymers for the Recognition of FLAG-Tagpeptide

**DOI:** 10.3390/ijms24043453

**Published:** 2023-02-09

**Authors:** Elsa Lafuente-González, Miriam Guadaño-Sánchez, Idoia Urriza-Arsuaga, Javier Lucas Urraca

**Affiliations:** 1Independent Researcher, E-28040 Madrid, Spain; 2Chemical Optosensors and Applied Photochemistry Group (GSOLFA), Department of Analytical Chemistry, Facultad de Química, Universidad Complutense de Madrid, E-28040 Madrid, Spain; 3Independent Researcher, E-28040 Madrid, Spain

**Keywords:** molecularly imprinted polymers, FLAG-tag, protein purification methods, epitope approach, solid phase extraction

## Abstract

FLAG^®^ tag (DYKDDDDK) is a small epitope peptide employed for the purification of recombinant proteins such as immunoglobulins, cytokines, and gene regulatory proteins. It provides superior purity and recoveries of fused target proteins when compared to the commonly used His-tag. Nevertheless, the immunoaffinity-based adsorbents required for their isolation are far more expensive than the ligand-based affinity resin used in combination with the His-tag. In order to overcome this limitation we report herein the development of molecularly imprinted polymers (MIPs) selective to the FLAG^®^ tag. The polymers were prepared by the epitope imprinting approach using a four amino acids peptide, DYKD, including part of the FLAG^®^ sequence as template molecule. Different kinds of magnetic polymers were synthesised in aqueous and organic media also using different sizes of magnetite core nanoparticles. The synthesised polymers were used as solid phase extraction materials with excellent recoveries and high specificity for both peptides. The magnetic properties of the polymers confer a new, effective, simple, and fast method in the purification using FLAG^®^ tag.

## 1. Introduction

Peptides are naturally occurring biological polymers formed by short chains of amino acid monomers (aa) linked by peptide bonds. They are distinguished from proteins based on size, usually containing approximately 50 or fewer amino acids. Peptides can be found isolated or as part of a huge protein in all living organisms, and their function is dependent on the types of amino acids involved in the chain and their sequence, as well as the specific shape of the peptide [1].

Protein detection and purification usually relies upon specific antibodies. Although there are many antibodies available commercially, they do not cover all proteins, especially if the protein has novel or unknown sequences and are time-consuming and expensive to produce. Epitope tags (also known as fusion tags or affinity tags) are a great alternative, as they can act as universal epitopes for detection and purification without disturbing the structure of the protein to which they are fused [2]. Epitope tagging is a technique in which a known epitope with a known antibody is fused to a recombinant protein by means of genetic engineering, allowing detection proteins for which no antibody is available [3]. The most used epitope tag is FLAG^®^ (DYKDDDDK), which consists of eight amino acids, including an enterokinase cleavage site.

The addition of a marker to the protein should not interfere with the native folding of proteins to which it is attached. The peptide sequence marker must be water soluble and must allow the surface exposure of the protein, so that it can easily interact with its ligand. It must also be suitable for a purification process by gentle affinity and low cost. The easy removal of the marker peptide, leading to a native product, is also advantageous. Due to the small size, the marker peptide may be encoded by a single synthetic oligonucleotide. Thereby, the FLAG-Tag^®^ marker is useful for the identification and purification of proteins. Elution of the protein can already be achieved by affinity chromatography, mediated by antibodies dependently on calcium, lowering the pH, or by competitive elution with synthetic peptides. Despite highly selective binding capacity, it is expensive. Besides the cost, they have several disadvantages: ligand leakage, instability, and the need for validation of antibody production. The stability of the affinity chromatography column depends on the nature and source of the extracts. Despite these drawbacks, the FLAG-Tag^®^ is useful in research and development. Several antibodies against this peptide have been developed to bind it in the presence of divalent metal cations, preferably Ca^2+^. FLAG^®^ proteins can be easily purified by ELISA or any other method of immunochemical purification, thus facilitating the characterization of the protein desired [4].

Molecularly imprinted polymers (MIP) are polymers with molecular recognition abilities, provided by the presence of a template molecule during their synthesis, and are excellent materials with high selectivity for sample preparation [5]. Therefore, molecularly imprinted polymers (MIPs) essentially act as artificial antibodies and are known and applied as antibody mimics for different applications, as excellent molecular recognition properties can be achieved with MIPs for a variety of molecules [6]. Rachkov and Minoura [7] proposed an epitope imprinting approach where only a small portion of the target protein is used to create a selective site. This bioinspired approach makes use of short peptide sequences as templates to generate binding sites capable of recognizing larger peptides. The necessity of using a large macromolecule as a template is thus overcome, although when used to bind the native protein, the problem of hindered diffusion to the sites may still be present unless the epitope is well exposed [8]. MIPs can be prepared in different physical formats [9] (bulk, thin films, spherical particles, etc.) and can be used in a wide variety of applications [10] (synthesis and catalysis, detection [11], sensors [12], solid phase extraction (SPE) [13], analytical separation [14], purification [15], etc.).

Magnetic nanoparticles, especially those based on Fe_3_O_4_, are the most often used substrate because of their ease of operation, surface modification and operation, excellent dispersibility in aqueous solution, good recoverability, and low toxicity [16]. Thus, magnetic MIPs (mMIPs) are synthesised by grafting MIP films onto the surface of modified Fe_3_O_4_ nanoparticles. They are capable of selectively recognizing and capturing the target analyte in a sample solution, and can be easily isolated from the solution by applying an external magnetic field [17]. Magnetite nanospheres [18] are aggregates of small nanoparticles, and also show superparamagnetic behaviour, with a size between 80 and 400 nm [19]. Thus, this feature makes nanospheres more convenient for their application in solution separations [20].

There are different types of core-shell mMIPs depending on the synthetic approach used for their preparation. The core-shell magnetic microspheres-MIP in which the MIP coating is grafted on the magnetic core surface. This is the simplest mMIP synthetic method [21]. The grafted magnetite nanoparticles in silica-MIP are silica nanoparticles which are synthesized by a conventional Stöber process, and the magnetite nanoparticles are grafted into de silica using a coprecipitation method. The MIP shell is grown afterwards around the magnetic nanoparticles [22]. In the last approach, the core-shell magnetic microspheres-silica-MIP is produced, in which a core of magnetite nanoparticles is synthesized first, a silica layer is grown around. Then, the magnetic-silica particles are functionalized with polymerizable groups at their surface, and a MIP coating is grafted on the surface [23].

Previous works have already demonstrated the ability and application of MIPs for the successful purification of the FLAG-tag peptide [18,20]. Nevertheless its application required improvements due the moderate reusability of the cartridges for SPE (25 times), high consuming time, and physical problems (blockages and cleaning). In this work is presented the synthesis of two mMIPs targeted against FLAG^®^-tag and its application as SPE sorbent materials, solving all the SPE conventional problems.

## 2. Results and Discussion

### 2.1. Chromatographic Evaluation

The development of an HPLC chromatographic method for the separation of DYKD and DYKDDDDDK peptides was carried out. For this purpose, a gradient method (Table 1) was employed, in which a binary AcN-TFA mixture (0.1% *v*/*v*) and a H_2_O-TFA mixture (0.1% *v*/*v*) were used as mobile phase.

As shown in the chromatogram in Figure 1, obtained after analysis of a standard solution of a mixture of DYKD and DYKDDDDK peptides with a final concentration of 20 mg∙L^−1^, the retention times obtained were 3.87 min (*W*_1/2_ = 0.124) for the DYKD peptide and 6.43 min (*W*_1/2_ = 0.105) for the DYKDDDDK peptide. In addition, calibration lines were plotted for both peptides over a concentration range of 0.5–20 mg∙L^−1^, yielding linear correlation coefficients greater than 0.98 and a dead time (*t*_0_) of 0.79 min. Column resolution (*Rs*), retention factors for both peptides (*k*), selectivity factor (*α*), and the number of plates forming the column (*N*) were then calculated (Table 2) from the following equations:(1)$$k=\frac{\left(t_{R}-t_{0}\right)}{t_{0}}$$
(2)N=5.54×[tRW12]2
(3)$$\propto =\frac{k_{B}}{k_{A}}$$
(4)Rs=2×[(tR)B−(tR)A](W12)A+(W12)B

To establish whether the separation of DYKD and DYKDDDDK peptides on the chromatographic column is optimal, the values obtained for the retention factor (k_DYKD_ = 3.90; k_DYKDDDDK_ = 7.14) should be between 1 and 5 (with values above this range not implying a significant increase in resolution), the selectivity factor (*α* = 1.83) higher than 1, and the resolution value (*Rs* = 22.35) higher than 1.5.

### 2.2. Synthesis of Magnetic Core-Shell Particles

The coating of magnetite nanoparticles with silica (Fe_3_O_4_@SiO_2_) has several advantages, such as effectively preventing the agglomeration of magnetite particles in aqueous solution, allowing a stable magnetic suspension to be obtained; easily coating various functional groups [24]; partially protecting iron oxide from acidic environments in which it would dissolve; and facilitating the self-assembly of molecules on the silica surface.

Scanning electron microscopy (TEM) was used to evaluate the dispersity, the isolation of the magnetic spheres, and the correct coating of these with silica for the eight synthesis methods studied in this work in order to determine the most optimal one. In method A (Figure 2), the correct formation of magnetic cores coated with a silica layer was observed, but with a wide diversity of sizes, irregular shapes, and an excess of silica, as the formation of silica spheres without a magnetic core was observed.

In method B (Figure 3) the formation of homogeneous, monodisperse, and perfectly isolated core-shell spheres was observed, but with a very small magnetic core size with respect to the silica layer, which greatly reduced the magnetic properties of the core-shell. Therefore, the same procedure was repeated, but with larger magnetic nanoparticles synthesized by the solvothermal method, obtaining core-shell nanospheres with a good silica coating, well differentiated, with an optimal size, and highly homogeneous. Nevertheless, the Fe_3_O_4_@SiO_2_ were not totally isolated but were attached to each other. In order to leave more space for reaction and avoid agglomeration of the core-shell microspheres, the synthesis was repeated, keeping the mass of magnetic nanoparticles constant and increasing the rest of the reagents and solvents by 18 times the initial amount.

In method C (Figure 4), it was seen that, for the core-shell nanoparticles with a co-precipitation magnetic core, the growth of silica spheres on the magnetic nanoparticles was not favoured. On the other hand, in the solvothermal magnetic cores, the formation of differentiated magnetic cores coated with silica was observed, but in an aggregated manner, preventing isolated coated spheres from being obtained. Therefore, we tried to dilute the solution of particles obtained with the intention of breaking the aggregates and separating the coated spheres, but no difference was observed.

Methods D, E, F, G, and H (Figure 5) did not prove to be effective procedures for obtaining core-shell particles, as in all cases irregular particle agglomerations were observed and the silica was mostly generated independently of the magnetic particles. As shown in Figure 3, the core-shell obtained after the reagent and solvent increment of method 2 proved to be the most suitable, as well-defined and isolated nanospheres were obtained, with a magnetic core in a suitable size range and a homogeneous silica coating. It was therefore selected for the synthesis of the molecularly imprinted magnetic polymers after surface functionalization.

SiO_2_ nanoparticles have proven to be one of the most favourable inorganic nanomaterials used as additives due to their low cost, good size control, and high surface reactivity. On the other hand, pure inorganic nanomaterials aggregate in the polymer matrix due to their large specific surface area and reactivity, leading to the formation of large defects and decreased film selectivity. Therefore, it is necessary to modify the surface area of nanomaterials to improve their dispersity and miscibility in the polymer matrix. For the functionalization by RAFT polymerization (Reversible Addition, Fragmentation, and Transfer Polymerization), the RAFT agent used was cumyldithiobenzoate, which is generated from the reaction of phenylmagnesium bromide with carbon disulphide. A silane was previously adhered to the silica surface through the -OH groups on the surface, which reacts with the RAFT agent, allowing polymerization. Finally, the synthesis of the polymers (mMIPs and mNIPs) was carried out in organic (after application of RAFT) and aqueous (after functionalization with double bonding agent) media.

### 2.3. mMIPs Characterisation

#### 2.3.1. X-ray Diffraction (XRD)

Figure 6 compares the XRD patterns of the magnetic nanoparticles before and after the polymerization process, which allows to confirm whether the polymerization conditions altered the structure of the magnetic cores. The diffraction pattern of the magnetic cores before polymerization (red line) showed diffraction maxima indexed to 220, 311, 400, 422, 440, and 511 reflections, typical of the cubic inverse spinel structure of magnetite (Fe_3_O_4_) or maghemite (γ-Fe_2_O_3_). In the diffraction pattern of the polymer-coated magnetic cores (blue line), it was observed that the peaks after polymerization are less intense and less defined due to the presence of the polymer layer surrounding the nanoparticles. The composition of free nanoparticles and MIP-coated nanoparticles was studied by calculating the lattice parameter a, which reflects the symmetry of the synthesized material by locating equivalent positions. For the calculation and comparison of the lattice parameter, the crystalline system was considered, which relates the distance between the crystal lattice planes (hkl) and the lattice parameters (*a*, *b* and *c*) by means of the following equation:(5)$$\frac{1}{d^{2}}=\frac{h^{2}+k^{2}+l^{2}}{a^{2}}$$
where *a* = *b* = *c* and *α* = *β* = *γ* = 90 °C for a cubic system. Values *a* (magnetite) = 0.838 ± 0.002 nm and *a* (mMIP) = 0.836 ± 0.002 nm were obtained for the magnetite cores before and after polymer coating, respectively, being very close to the literature values for magnetite (*a* = 0.8396 nm) and maghemite (*a* = 0.8352 nm), indicating therefore that a mixture of both was in the core-shell.

Finally, the average crystal size of the magnetic cores and mMIPs was calculated using Scherrer’s formula [25]:(6)$$D=\frac{K\lambda }{\beta \cos\theta }$$
where *K* is a dimensionless parameter that depends on the packing and whose value is close to unity (usually 0.9), *β* is the half-height width of the diffraction peak, *θ* is the Bragg angle, and *λ* is the wavelength of the X-rays used in the experiments, whose value is 1.540598 Å. The *D* values obtained for the magnetic nanoparticles before and after the polymerization process were 10.8 nm and 12.1 nm, respectively. In conclusion, the results obtained indicated that the magnetic nanoparticles are incorporated into the polymer (MIP/NIP), and their structure does not undergo a significant change after the polymerization process.

#### 2.3.2. Fourier Transform Infrared Spectrometry (FTIR)

To verify the presence of polymer coating onto the core-shell magnetic nanoparticles, the spectra of the Fe_3_O_4_ and Fe_3_O_4_@SiO_2_ nanoparticles and the synthesized polymers were measured in both aqueous and organic media (Figure 7 and Figure 8). In the FTIR of the magnetic magnetite nanoparticles (red line), the 583.35 cm^−1^ band was assigned to the vibration of the Fe-O bonds of Fe_3_O_4_, being the most representative band of magnetite, and the broad band at 3442.75–3423.35 cm^−1^ was assigned to the presence of -OH groups on the surface. In the spectrum of the core-shell nanoparticles, a decrease in the Fe-O vibrational band was observed due to the coating of the magnetic particles with silica, a band at 800.75 cm^−1^ was observed, which was assigned to the vibration of the Si-O bonds, another band was observed at 965.75 cm^−1^, which was attributed to the vibration of the Si-O-H bonds, and a band at 1091.73 cm^−1^, which was assigned to the vibration of the Si-O-Si bonds. In the FTIR of the organic (green line) and aqueous (blue line) mMIPs, a band at 2922.27 cm^−1^/2922.01 cm^−1^ was observed, which was attributed to the presence of C-H stress vibrations of the -CH_3_ and -CH_2_ groups of the formed polymer layer, and the decrease in the bands is attributed to the vibrations of the Si-O, Si-O-H, and Si-O-Si bonds when the synthesis of the mMIPs on the magnetic core-shell particles is achieved. This confirms that the nanoparticles are embedded in the polymeric material.

#### 2.3.3. Transmission Electron Microscopy (TEM)

Figure 8A shows the transmission micrograph of the magnetic particles (Fe_3_O_4_) synthesized by the solvothermal method, which was finally selected for the core-shell synthesis because the nanoparticles have a high homogeneity, and the particle size proved to be the most suitable for the correct coating with silica, as seen above. Figure 8B shows the mMIPs synthesized in both organic and aqueous media, confirming in both cases the correct coating of the core-shell magnetic particles by the molecular imprinted polymer (MIP) in the polymerization process. Thereby, particles were obtained with an average diameter of 88 ± 18 nm and of 104 ± 19 nm for the mMIPS synthesized in organic and aqueous media, respectively.

### 2.4. mMISPE Procedure

As it was described by Gómez-Arribas et al. [13,15] previously, EAMA monomer is going to interact through the amino groups with the acid carboxylic acids groups that FLAG bears in its structure. These kinds of interactions in aqueous media are electrostatic or ionic. In this way, depending on the pH of the media, the protonated/deprotonated functional groups of the functional monomers can interact with the protonated/deprotonated functional groups of the FLAG-tag. In the case of MAA as a functional monomer, this monomer can interact with the amine group of the N-terminal and also with the one of the lysine. In this case, the interaction is through H-bonding in acetonitrile as porogen. In the case of the EAMA monomer, the amine groups can interact with the carboxylic acid groups of the acids that bear the different amino acids in FLAG-tag. To carry out the solid-phase extraction process with the synthesized mMIP, the conditioning, loading, washing, and elution procedures were optimized. Conditioning was carried out in two steps, the first with MeOH to remove non-specific retained substances from the polymer, and the second with Tris buffer (20 mM) to condition the polymer to the medium in which the extraction was performed. Tris buffer was chosen as the solvent for the SPE process as it is the most commonly used buffer in the protein purification process due to the high stability of the peptides in it. After peptide loading, the peptides were washed with Tris (5 mM), which kept the pH stable (7.5), favouring the ionic interaction between the functional monomers and the target molecules, and preventing peptide denaturation, while, at the same time, eliminating the substances retained in the polymer in a non-specific manner. Finally, for the elution stage, TBA was used because it is capable of creating an ionic pair of higher affinity with the polymer that favours the elution of the analyte (peptides). In this way, a recovery study of the DYDK and DYKDDDDK peptides was carried out on both the polymers synthesized in aqueous and organic media (Table 3 and Table 4, respectively).

Both for the polymers synthesized in organic and aqueous media, it was observed that the recoveries were always higher in the mMIPS than in their corresponding mNIPs, which shows that the retention of the peptides by the imprinted polymers is highly specific and certifies the effectiveness of the imprinting process used in this work. On the other hand, it could be observed that the recoveries by mMIP in the DYKDDDDDK peptide (R(%)_aqueous_ = 105.6%; R(%)_organic_ = 67.1%) were always higher than those of the DYKD template molecule peptide (R(%)_aqueous_ = 79.1%; R(%)_organic_ = 51.6%), which may be due to higher non-specific retention towards the first peptide due to a higher number of amino acids in its sequence susceptible to interaction, and to the increased hydrophobicity of the peptide due to the higher number of carbon atoms in its chain. In the case of mNIP in organic medium, the DYKD peptide (R(%)_aqueous_ = 43.6%; R(%)_organic_ = 0%) was not retained, while the DYKDDDDK (R(%)_aqueous_ = 41.4%; R(%)_organic_ = 11.6%) was slightly retained in a non-specific way, and, in aqueous medium, both peptides were retained in a similar way.

Both polymers are useful as materials for solid phase extraction. The polymer made in aqueous media rendered better recoveries (close to 100% for all the concentrations studied), but the imprinting factor was not very high (2.5–2.8) for FLAG-tag. On the other hand, the polymer synthesized in acetonitrile offered better IF (4.8–7.2), which means higher specificity. Nevertheless, the recoveries obtained in this case are smaller for the mMIP, between 63% and 72%. In addition, a selectivity test was achieved for related (D) and non-related single amino acids (E) for a fixed concentration of 5 mg∙L^−1^ for all the tested molecules. As Figure 9 shows, no retention of these compounds was observed for the imprinted and the non-imprinted materials. This fact suggests the very important role of the size of the molecule for the retention in the cavities of the polymer. These molecules, even having imprinted functional groups of DYKD, are not retained at all, which reveals the imprinting factor of the synthesized materials. Finally, it should be pointed out that the mMIPs could be reused at least 80 times without noticeable loss in performance or reproducibility. This fact confirms the robustness of the imprinted material versus conventional anti-FLAG affinity gels, where in their specifications are recommended for less than five uses. 

### 2.5. Binding Isotherms

The binding properties and the homogeneity of the binding sites of the mMIPs were evaluated by equilibrium analysis. A rebinding test was carried out in Tris buffer (20 mM). Thus, a rebinding test using a rebinding molecule was used in a mixture of solvents AcN:H_2_O (pH = 3). The binding features of FLAG-tag imprinted and non-imprinted materials synthesized in aqueous and organic media were fitted using the Freundlich isotherm (FI) by plotting the experimental binding data (Figure 9) in log format:logB=loga+mlogF
where *B* and *F* represent the amounts of template adsorbed by the column and free in solution to the breakthrough volume, respectively. *a* is the binding capacity and *m* is the so called heterogeneity index. This latter parameter takes values from 1 to 0 and increases with decreasing heterogeneity of the material. According to Rampey et al. [26], from this equation, the affinity constant and the total number of binding sites can be calculated as well. 

The curves (Figure 10) show the higher affinity of the imprinted polymer over the non-imprinted polymer. The differences in the ADs of the evaluated polymers (Table 5) show that the binding capacity (a) and the total number of binding sites (N) of the MIP are higher than that for the corresponding NIPs. The total number of binding sites are N_MIP_: 43 ± 2 and 35 ± 2 (µmol g^−1^) and N_NIP_: 11 ± 3 and 9 ± 1 (µmol g^−1^). The same occurs for the weighted average affinities (K_MIP_: 58 ± 3 and 44 ± 3 (mM^−1^), K_NIP_: 12 ± 1 and 11 ± 1 (mM^−1^)) that are also higher for the MIPs in the measured concentration range. In addition, the comparison of the heterogeneity parameter of the polymer reveals that the heterogeneity index (*m*) is smaller in the imprinted polymers (*m*: 0.65 ± 0.01) than in the corresponding NIPs (*m*: 0.94 ± 0.03 and 0.73 ± 0.02). In agreement with Rampey et al., the binding sites in imprinted polymers should be more heterogeneous than those in non-imprinted ones and then render smaller m values. Thus polymers synthesized in different media show similar behaviour for the retention of FLAG, but capacity factors are slightly higher in the case of the polymer synthesized in aqueous media.

## 3. Materials and Methods

### 3.1. Materials and Reagents

DYKD peptide and DYKDDDDK peptide FLAG^®^, Peptide Sciences (Henderson, NV, USA); Methacrylic acid (MAA), citric acid anhydrous with trisodium salt, trimethylolpropanetriacrylate (TRIM), iron(II) chloride tetrahydrate (FeCl_2_·4H_2_O), iron(III) chloride hexahydrate (FeCl_3_·6H_2_O), ethanol (EtOH), ethylene glycol, polyethylene glycol (PEG), sodium acetate, ammonium hydroxide (25%), phenylmagnesium bromide, triethylamine (Et_3_N), 3-(trimethoxysilyl)propyl methacrylate (98%), and tetraethyl orthosilicate (TEOS) were purchased from Sigma-Aldrich (Madrid, Spain); Trifluoroacetic acid (TFA), N,N′-Ethylenebisacrylamide (EBAA), and 4-(Chloromethyl)phenyl trichlorosilane (97%) were supplied by Alfa Aesar; Cyclohexane by GPR Rectapur; N-(2-aminoethyl)methacrylamide hydrochloride (EAMA) from Polisciences (Eppelheim, Germany); 2,2′-azobis(2,4-dimethylvaleronitrile (ABDV) as obtained from Wako Chemicals (Neuss, Germany); Dimethylformamide (DMF) was purchased from VWR Chemicals (Leuven, Belgium); Tetrabutylammoniumhydrogensulfate (98%) (TBA) and ammonium persulfate (APS) as supplied by Merck; Toluene by Panreac; Hexadecyltrimethylammonium bromide (CTAB) by Fluka (Buchs, Switzerland); Sodium hydroxide was obtained from Quimipur (Madrid, Spain); Carbon disulfide was supplied by AnalaRNormapur; Water (H_2_O) was purified through a Milli-Q system from Millipore (Bedford, MA, USA); Acetonitrile (AcN), dimethyl sulfoxide (DMSO) and methanol (MeOH) were provided by Fisher scientific.

### 3.2. Apparatus and Instrument

Analytical balance Metler AT261 Delta Range; pH-met ORION 710A; HP 1200 series chromatograph with a diode array detector from Agilent Technologies, (Palo Alto, CA, USA); Torricelli vacuum pump from Telstar (Madrid, Spain); Transmission electron microscopes (TEM) JEM 2100 HT y JEM 1400; FT-IR Tensor 27 Spectrometer with DlaTGS detector; VibraCell 72434 (Bioblock F-Scientific, Illkirch, France).

### 3.3. Synthesis of Magnetite Cores (Fe_3_O_4_)

The synthesis of magnetite nanoparticles (NPs) was carried out by three different methods. The first (method 1) was the solvothermal method, in which magnetic nanospheres were synthesized by a solvothermal reaction. For this purpose, 2.43 g of FeCl_3_·6H_2_O and 6.48 g of sodium acetate were dissolved in 72 mL of ethylene glycol in a glass vessel under magnetic stirring at room temperature. Then, 1.6 mL of PEG was added and magnetic stirring was maintained for 30 min. The reaction mixture was poured into two Teflon reactors and left in an oven at 190 °C for 24 h. Finally, with the help of a magnet, the obtained magnetic particles were washed with portions of methanol (2 × 250 mL) and water (2 × 250 mL). They were placed in an oven at 50 °C until completely dry.

The second (method 2) method used was coprecipitation [27], for which 0.86 g of FeCl_2_·4H_2_O and 2.36 g of FeCl_3_·6H_2_O were added to a two-hole round-bottom flask and dissolved in 40 mL of Milli-Q water. The mixture was refluxed at 80 °C in argon atmosphere with magnetic stirring at 900 rpm and 5 mL NH_4_OH was added. After 30 min, 1.34 g of anhydrous citric acid with trisodium salt previously dissolved in 2 mL of water was added dropwise, and the reaction was maintained for 90 min. The reaction mixture was allowed to cool, centrifuged to isolate the magnetic particles, and the magnetic NPs were washed several times with water. 

In the last method (method 3), stabilized nanoparticles of magnetite were prepared by thermal decomposition of precursors in a high boiling point solvent, using oleic acid (OA), in the presence of oleylamine (OLA), as stabilizing agent [28]. Iron (III) acetilacetonate (Fe(acac)_3_) hexahydrate was used as iron precursor. Biphenylether (PE) and 1,2-hexadecanediol (HDD) acted as solvent and reducing agent, respectively. In each case, the reactant, in a 10:7:6:6:3 molar ratio of HDD:PE:OA:OLA:Fe(acac)_3_, were added to a three-neck round-flask, and argon flux was passed through the flask. After 10 min under argon flux and stirring, the temperature was increased up to 200 °C, and the mixture was treated in such conditions for 60 min. Then, the temperature was increased again up to 265 °C, and the mixture was treated for 60 min. The mixture was let down to reach room temperature, washed with ethanol, centrifuged, and redispersed in hexane. The process was repeated until the IR spectra indicated that no free oleic acid molecules were retained in the sample.

### 3.4. Silica Coating Methods (Fe_3_O_4_@SiO_2_)

After obtaining the magnetic nanoparticles, a first coating with silica, SiO_2_, was carried out, obtaining core-shell magnetic nanoparticles, Fe_3_O_4_@SiO_2_. Several SiO_2_ coating methods were tested for this synthesis. The first synthesis method (method A) used was a sol-gel process, in which 300 µL of the magnetic NPs (method 3) were dissolved in a mixture of EtOH (16 mL), H_2_O (4 mL), and hexane (700 µL), and sonicated with an ultrasound probe for 15 min. Finally, 1.33 mL NH_4_OH (25%) and 0.2 mL TEOS were added and sonicated for another 2 h.

For the second method [29] (method B), an aliquot of 0.5 mL of a dispersion of magnetic NPs (methods 1 and 3) in chloroform (6.7 mg Fe∙mL^−1^) was taken and added over 5 mL of a 0.055 M aqueous CTAB solution. The mixture was stirred vigorously at room temperature for 30 min, then stirred at 60 °C for 10 min to evaporate the chloroform, and the resulting mixture was added over a previously prepared solution of H_2_O (45 mL) and 2M NaOH (0.3 mL). The mixture was heated to 70 °C, and 0.5 mL TEOS and 3 mL ethyl acetate were added and under shaking for 10 min. Subsequently, a further 50 µL of TEOS was added and left to stir for 3 h. The product obtained was first washed with water (1 × 100 mL) and EtOH (3 × 100 mL) and then with HCl (1 × 40 mL) at 60 °C with stirring for 3 h to remove the traces of CTAB.

For the third mode of core-shell synthesis [30] (method C), 0.10 g of nanoparticles from the solvothermal and coprecipitation methods (methods 1 and 2) were mixed with 18 mL of 0.1M HCl and placed in the ultrasonic bath for 10 min. Magnetic NPs were isolated with the help of a magnet, washed with water (3 × 50 mL), and dispersed in a previously prepared mixture of EtOH (16 mL), H_2_O (4 mL), and NH_4_OH (28%) (200 µL). 30 µL of TEOS was added and placed under shaking at room temperature for 6 h. The obtained nanospheres were isolated with the help of a magnet, washed with ethanol and water, and dispersed by sonication and mechanical stirring in a mixture of CTAB (60 mg), H_2_O (16 mL), NH_4_OH (28%) (228 µL), and EtOH (2 mL). 100 µL of TEOS was added under continuous stirring, allowed to react again for 6 h, and washed again with EtOH and water. The obtained nanospheres were redispersed in a solution of EtOH (65 mL) and NH_4_NO_3_ (0.13 g), and the mixture was stirred at 60 °C for 10 min to remove the CTAB; finally, the product was isolated with a magnet and washed with ethanol.

For the fourth synthesis route [31] (method D), a volume of 2 mL aliquot of an aqueous dispersion of magnetic NPs from the co-precipitation method (method 2, 20 mg Fe mL^−1^) was taken and added over a mixture of EtOH (8 mL) and NH_4_OH (25%) (142 µL), stirred for 15 min at 30 °C, and then 114 µL TEOS was added over 6 h (20 µL TEOS/h). The resulting nanospheres were isolated with a magnet and washed with ethanol and water repeatedly.

For the fifth procedure [32] (method E), a 0.3 mL aliquot of the aqueous dispersion of the magnetic particles from the co-precipitation method was taken (method 2), added to a solution containing NaCl (0.58 g), H_2_O (40 mL), and EtOH (16 mL), and stirred for 30 min at 35 °C. Finally, 222 µL of TEOS was added and stirred for 24 h at the same temperature.

For the sixth method [33] (method F), 0.23 g of PEG 6000 was dispersed in 4.8 mL of cyclohexane by sonication and 0.2 mL of a dispersion of magnetic nanoparticles (method 3) in cyclohexane (150 mg·mL^−1^) and 50 µL of NH_4_OH were added. The mixture was stirred for 1 h at room temperature and 50 µL of TEOS were added, stirred for 24 h, and the resulting product was washed with ethanol (3 × 100 mL).

In the penultimate synthesis mode [34] (method G), 0.3 g of silica powder was added to a mixture of 50 mL H_2_O and 2 mL of the aqueous dispersion of the magnetic particles from the co-precipitation (method 2) under vigorous stirring at 85 °C for 30 min. Then 50 mL of an aqueous NH_4_OH solution (1.0% *v*/*v*) were added, and the reaction was kept under stirring at the same temperature for 2 h. Silica residues were removed by filtration (0.8 µm filter).

For the last synthesis method [35] (method H), 5 mg of the particles from the solvothermal and coprecipitation methods (methods 1 and 2) were dispersed in a mixture of EtOH (16 mL), H_2_O (4 mL), and NH_4_OH (25%) (224 µL) and sonicated for 20 min. Then 100 µL of TEOS were added, and the reaction was left to stir at 30 °C for 6 h; the product obtained was washed with ethanol (3 × 50 mL) and water repeatedly (3 × 50 mL).

### 3.5. Functionalization of the Surface

In order to carry out the polymerization process around the core-shell particles, a previous surface functionalization step was carried out. The first approach consisted of modifying the Fe_3_O_4_@SiO_2_ materials by grafting a polymer onto their surface by means of RAFT (Reversible Addition Fragmentation Chain Transfer) polymerization [36]. For this purpose, 25 mg of the magnetic core-shell particles were suspended in 10 mL of anhydrous toluene in a 25 mL two neck round-bottom flask, and the mixture was purged with argon for 10 min. Then 10 µL of 4-(chloromethyl)phenyltrichlorosilane and a mixture of triethylamine (10 µL) and anhydrous toluene (1 mL) were added dropwise; the reaction was refluxed for 24 h at 800 rpm. The product obtained was centrifuged, washed with toluene and methanol (3 × 10 mL), and left to dry at 45 °C overnight. The next day, 10 mL of anhydrous toluene was added to a new 25 mL two-neck round-bottom flask and purged with argon for 15 min. Next, 310 µL of phenylmagnesium bromide and 47 µL of carbon disulfide were added and left to react under reflux for 1 h at 50 °C and 600 rpm. The silanization product (20.70 mg) obtained in the first synthesis was added and allowed to react for 90 min at 50 °C and 800 rpm. The final product was centrifuged, washed with toluene, methanol, and acetone (3 × 50 mL), and left to dry at 45 °C overnight. The second approach was to use an agent that bears a double bond in its structure, allowing an anchorage to the surface facilitating the subsequent polymerization reaction. For this purpose, 180 mg of core-shell magnetic particles were dispersed in 20 mL of toluene and 1.5 mL of 3-(trimethoxysilyl) propyl methacrylate (double bonding agent), and 120 µL of Et_3_N were added. The mixture was kept stirred at 45 °C for 24 h.

### 3.6. Magnetic MIP Synthesis (mMIPs)

The synthesis of magnetic molecular imprinted polymers (MIPs) was carried out using two different polymerization methods depending on the type of functionalization previously employed. A MIP was synthesized in an organic medium after application of the RAFT method, and another different MIP was obtained in an aqueous medium after functionalization with the double bonding agent. For the polymer in organic medium, the molar ratio between the template molecule (DYKD peptide), the functional monomer (EAMA), and the crosslinking agent (TRIM) was 1:4:20, respectively. Thus, 0.9 mg of DYKD (Figure 11) was dissolved in a 20 mL mixture of an AcN (with/DMF solution (9:1 *v*/*v*)) and 1 mg of EAMA. Next, 2.34 mg of the magnetic core-shell particles obtained after the RAFT process and 10 µL TRIM were added. It was sonicated for 15 min with the ultrasound probe, 1.1 mg ABDV was added and purged with nitrogen for 5 min. The pre-polymerization mixture was placed in a roller stirring oven at 55 °C for 24 h to allow the polymerization process to take place. For the extraction of the template molecule, the polymer obtained was washed with successive portions of MeOH, 1% MeOH/TFA, and again MeOH to remove the TFA residues. The same procedure was followed to obtain the non-imprinted polymer (mNIP) but without the use of the template molecule. In the aqueous polymer, the molar ratio between the template molecule (DYKD peptide), the functional monomer (MAA), and the crosslinking agent (EBAA) was also 1:4:20, respectively. Thus, 1.4 mg of DYKD was dissolved in 50 mL H_2_O and 100 µL DMSO; 8 µL MAA, 10 mg of the magnetic core-shell particles obtained after functionalization with the double-bonding agent, and 43.6 mg EBAA were added. It was sonicated 15 min with the ultrasound probe; 60 mg of APS was added and purged with nitrogen for 5 min. The thermal polymerization process and extraction of the template molecule were carried out under the same conditions as those used for the organic mMIP. The same procedure was followed to obtain the non-imprinted polymer (mNIP), but without the template molecule.

### 3.7. Chromatographic Evaluation

For the chromatographic evaluation of the polymers, an Agilent Technologies HP 1200 chromatograph equipped with a binary pump, an in-line degasser, an autosampler, an automatic injector, a column thermostatiser with a temperature range from 10 °C below ambient temperature to 80 °C (±0.8 °C), and a diode array detector (DAD) as a detection system were used. For the analysis of peptides (DYKD and DYKDDDDK), an ACE Excel 2 C18-PFP column (100 × 2.1 mm, 2 µm) was used, and a gradient mode method was applied using a binary mixture of AcN-TFA (0.1% *v*/*v*) as mobile phase. The injection volume was 5 µL while maintaining the temperature at 45 °C, and the absorption wavelength was set at 210 nm. The flow rate was kept constant at 0.4 mL min^−1^.

### 3.8. Solid Phase Extraction Process (mMISPE)

The solid phase extraction process for both the polymers synthesized in organic and aqueous media consisted of five steps: 2 conditioning, loading, washing and, elution. For the first conditioning, 20 mg of mMIP/mNIP was weighed and 5 mL of MeOH was added. The resulting suspension was shaken for 5 min and the supernatant was removed with the help of the magnet. In the second conditioning, 5 mL of a 20 mMTris buffer solution (pH = 7.5) was added, shaken for 5 min, and the supernatant was removed with the aid of the magnet. For loading, 1 mL of a peptide solution at a given concentration (5, 10, and 20 mg∙mL^−1^) was added, shaken for 5 min, and the supernatant liquid was decanted with the help of a magnet. In the wash, 3 mL of the 5 mM buffer solution was added to remove the retained non-specific compounds, shaken for 30 s, and the supernatant liquid was again removed. Finally, in the elution step, 1 mL of an aqueous TBA solution (1% *v*/*v*) was added, shaken for 5 min, and the supernatant liquid was collected in an HPLC vial.

### 3.9. Equilibrium Rebinding Experiments

mMIP/mNIPs (10 mg) were weighed individually into 2 mL glass vials for mixing with 1 mL of 20 mM Tris buffer solution containing increasing concentrations of DYKDDDDK over the range 0.005–2.0 mM and stirred at room temperature in the dark for 24 h. The concentration of DYKDDDDK was determined by HPLC-DAD as described above. The amount of polymer-bound analyte (B) was calculated by subtracting non-bound analyte (F) from the initial analyte concentration in the mixture. The amount of DYKDDDDK bound to the polymer (B) was calculated by subtracting the free amount of the former (F) from the initial DYKDDDDK concentration in the mixture. Binding experiments were achieved in duplicate.

## 4. Conclusions

As we have seen in this work, the development of DYKD and DYKDDDDK (FLAG^®^) selective molecular imprinted polymers of magnetic character has been achieved following the “epitope approach” as a synthetic strategy. The synthesis of silica-coated magnetite core-shell nanoparticles with spherical shape and high magnetic capacities has been achieved, making these particles ideal to act as polymerization support. The molecular imprinted polymers synthesized in this work, in both aqueous and organic media, have yielded very optimal results and are therefore both suitable as MIPs as sorbent materials in solid phase extraction. Finally, the proposed mMISPE method has proved to be fast, easy to apply thanks to the magnetic separation (less than 20 min per sample), and highly reusable (more than 80 times), being considered as a promising alternative for the determination and purification of peptides and proteins.

## Figures and Tables

**Figure 1 ijms-24-03453-f001:**
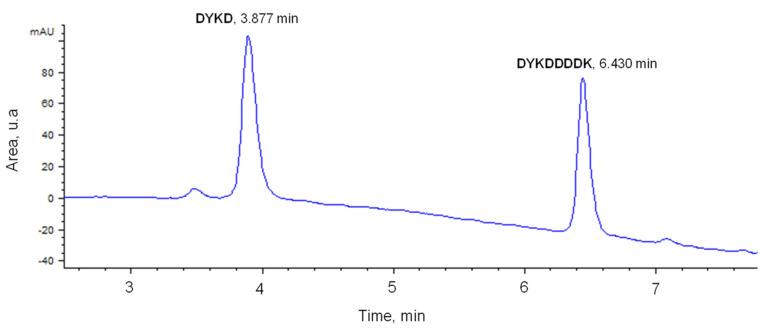
Chromatogram obtained from the analysis of a standard solution of the peptides DYKDD and DYKDDDDDDK (20 mg∙L^−1^).

**Figure 2 ijms-24-03453-f002:**
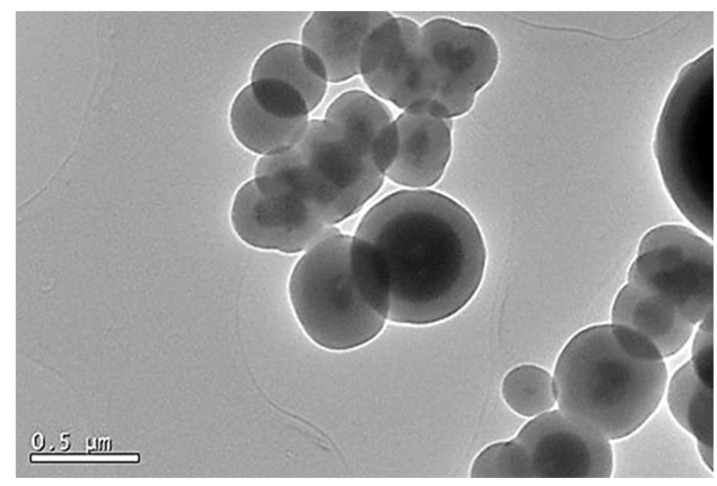
Transmission micrographs of Fe_3_O_4_@SiO_2_ nanoparticles (Method A).

**Figure 3 ijms-24-03453-f003:**
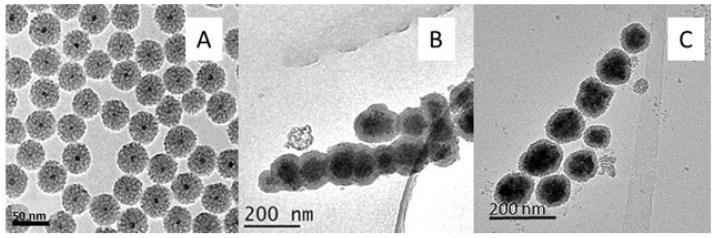
Transmission micrographs of Fe_3_O_4_@SiO_2_ nanoparticles (Method B); (**A**) magnetic nanoparticles 10 nm (Method 3); (**B**) magnetic nanoparticles via solvothermal (Method 1); and (**C**) magnetic nanoparticles via solvothermal (Method 2) with 18-fold increase in reagents and solvents.

**Figure 4 ijms-24-03453-f004:**
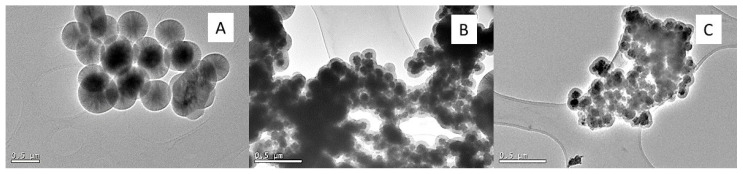
Transmission micrographs of Fe_3_O_4_@SiO_2_ method C; (**A**) magnetic core coprecipitation; (**B**) magnetic core via solvothermal; (**C**) magnetic core dilution via solvothermal.

**Figure 5 ijms-24-03453-f005:**
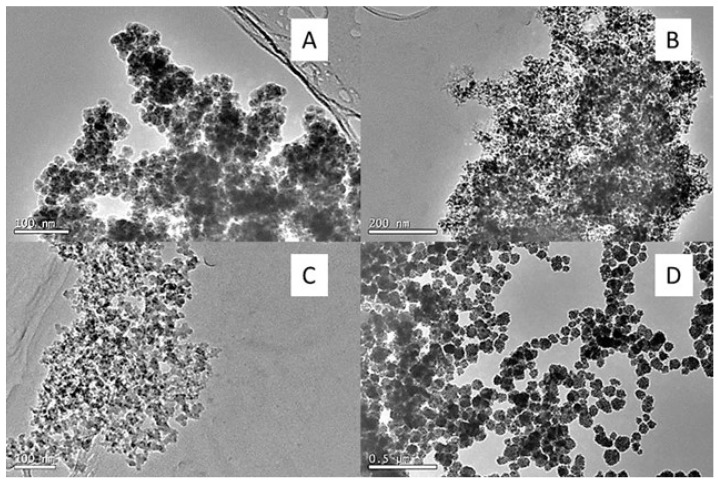
Transmission micrographs of Fe_3_O_4_@SiO_2_; (**A**) and (**B**) method D, E; (**C**) method G; (**D**) method H.

**Figure 6 ijms-24-03453-f006:**
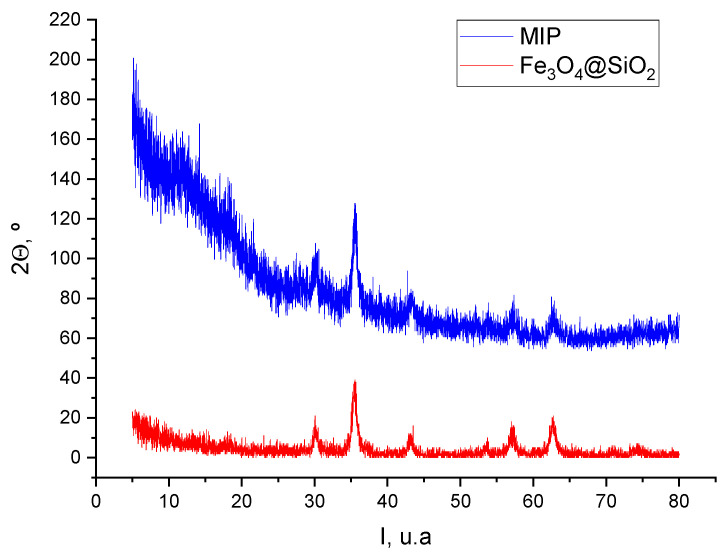
X-ray diffraction spectra of magnetic nanoparticles before (red line) and after (blue line) the polymerization process.

**Figure 7 ijms-24-03453-f007:**
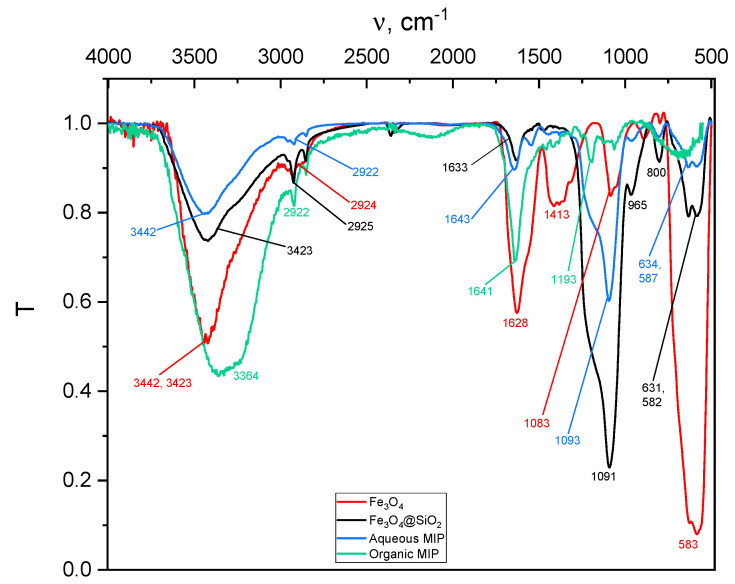
FT-IR spectra of the magnetic particles, Fe_3_O_4_, (red line), the magnetic core-shell particles, Fe_3_O_4_@SiO_2_, (black line), mMIP synthesised in aqueous medium (blue line), and mMIP synthesised in aqueous medium (green line).

**Figure 8 ijms-24-03453-f008:**
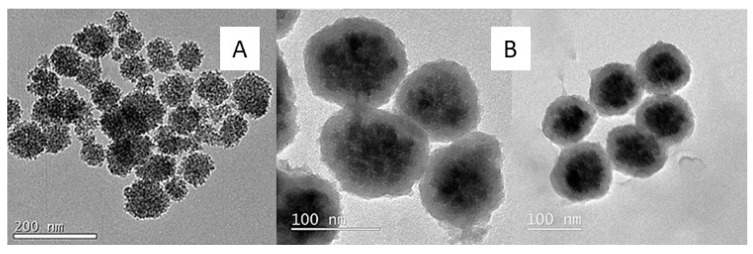
(**A**) Transmission micrographs of Fe_3_O_4_ nanoparticles synthesized by the solvothermal method. (**B**) Transmission micrographs of mMIP in organic medium (**left**) and mMIP in aqueous medium (**right**).

**Figure 9 ijms-24-03453-f009:**
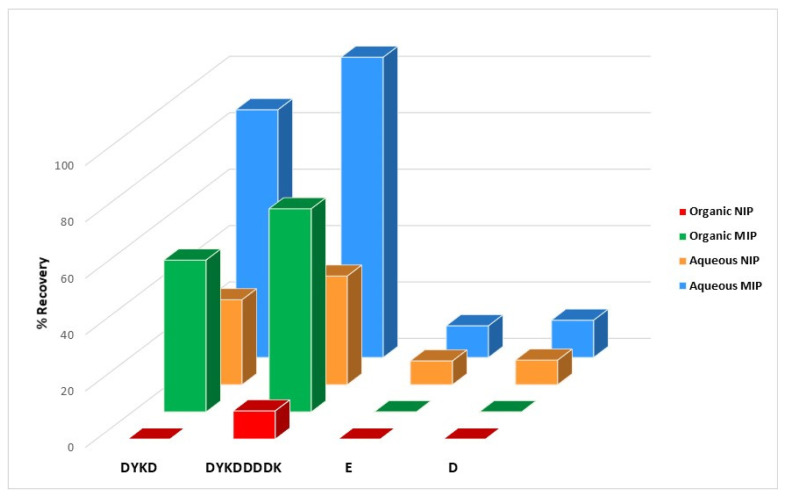
Recoveries (%) obtained after elution for DYKD, DYKDDDDK, E, and D compounds for a fixed concentration of 5 mg∙L^−1^ each following the whole MISPE process.

**Figure 10 ijms-24-03453-f010:**
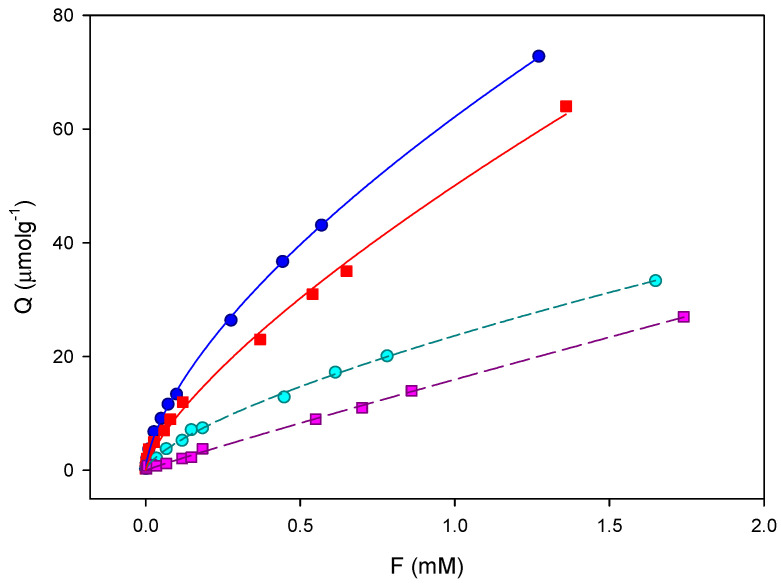
Equilibrium binding isotherms for DYKDDDDK to the mMIP (blue) and to the corresponding NIP (cyan), synthesised in aqueous solvent, and the mMIP (red) and to the corresponding NIP (pink), synthesised in organic solvent. The solid lines correspond to the fits to the Freundlich equation (B = aF^m^) (10 points, n = 2).

**Figure 11 ijms-24-03453-f011:**
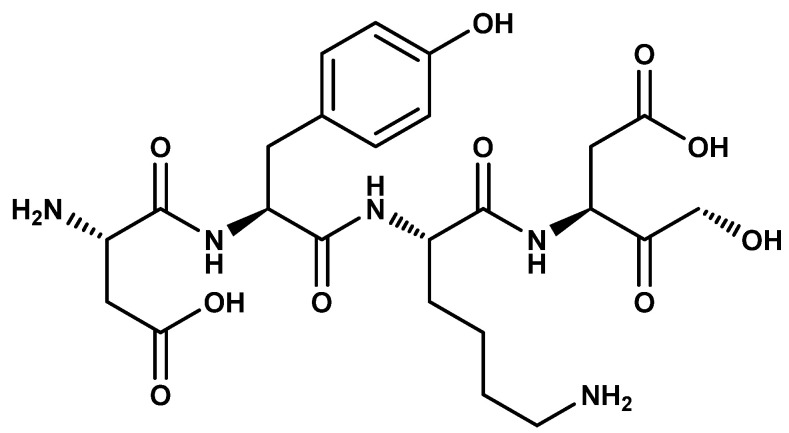
Scheme DYKD peptide used as template molecule.

**Table 1 ijms-24-03453-t001:** Gradient conditions used for chromatographic separation of peptides.

Time (min)	% AcN/TFA	% H_2_O/TFA
0–9	1	99
9.0–9.1	8	92
9.1–17	70	30
17–25	1	99

**Table 2 ijms-24-03453-t002:** Chromatographic parameters calculated from the standard solutions of DYKDDD and DYKDDDDDDDDDDK peptides (20 mg∙L^−1^).

Peptide	*k*	*N*	*α*	*Rs*
DYKD	3.90	5413.65	1.83	22.35
DYKDDDDK	7.14	20657.37		

**Table 3 ijms-24-03453-t003:** Recovery rates (R, %), relative standard deviations (RSD, %) and imprinting factors (IF) of the peptides in the polymers synthesized in aqueous media.

Loading Solution (mg·L^−1^)	DYKD					DYKDDDDK				
	R (%) mMIP	RSD (%) mMIP	R (%) mNIP	RSD (%) mNIP	IF	R (%) mMIP	RSD (%) mMIP	R (%) mNIP	RSD (%) mNIP	IF
5	87.8	4.7	30.1	4.1	2.9	106.4	4.0	38.5	4.4	2.8
10	85.7	1.5	50.8	1.1	1.7	107.3	1.6	44.5	0.6	2.4
20	63.9	3.4	50.0	2.0	1.3	103.0	2.4	41.2	1.4	2.5

**Table 4 ijms-24-03453-t004:** Recovery rates (R, %), relative standard deviations (RSD, %) and imprinting factors (IF) of the peptides in the polymers synthesized in organic media.

Loading Solution (mg·L^−1^)	DYKD					DYKDDDDK				
	R (%) mMIP	RSD (%) mMIP	R (%) mNIP	RSD (%) mNIP	IF	R (%) mMIP	RSD (%) mMIP	R (%) mNIP	RSD (%) mNIP	IF
5	53.8	3.9	0.0	0.0	N.A.	72.0	8.0	10.0	0.6	7.2
10	49.9	1.7	0.0	0.0	N.A.	66.1	0.3	11.8	0.4	5.6
20	51.2	0.1	0.0	0.0	N.A.	63.4	0.2	13.0	1.2	4.8

N.A.: Not applicable.

**Table 5 ijms-24-03453-t005:** Equilibrium binding isotherm parameters for the uptake of FLAG-tag (in Tris buffer 20 mM) synthesized by their corresponding MIPs and the NIPs in aqueous and organic solvent. The polymer binding capacity (*a*) and the binding site heterogeneity index (*m*) were obtained from the fit of the experimental data to the corresponding Freundlich isotherm.

Polymer	*K*_K_ (mM⁻^1^)	*N_K_*_1__–_*_K_*_2_ (µmol g⁻^1^)	*m*	*a* (µmol g⁻^1^ (mM⁻^1^)*^m^*)	*r* ^2^
MIP (aq)	58 ± 3	43 ± 2	0.65 ± 0.01	62 ± 1	0.9998
NIP (aq)	12 ± 1	11 ± 3	0.94 ± 0.03	15 ± 1	0.980
MIP (org)	44 ± 3	35 ± 2	0.65 ± 0.01	50 ± 1	0.980
NIP (org)	11 ± 1	9 ± 1	0.73 ± 0.02	13± 1	0.997

## Data Availability

The data presented in this study are available on request from the corresponding authors.

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
