# Peer review of "Core-Shell Magnetic Imprinted Polymers for the Recognition of FLAG-Tagpeptide"

_ijms, 2023, doi:10.3390/ijms24043453_

Round 1

Reviewer 1 Report

Attached in the word file.

Reviewer 2 Report

The manuscript describes the development of a core-shell magnetic imprinted polymer (mMIP) for the purification of FLAG-tagged proteins. Unfortunately, the title of the manuscript is not harmonized with its content and the defined aim, as there are still relevant experiments to be done in order to demonstrate the fit-for-purpose and actual functionality of the mMIP. Only then the scientific significance and impact of the manuscript may justify its publication in International Journal of Molecular Sciences.

Further comments and observations are presented or reiterated in more detail below:

  1. Introduction must be more focused - eliminate many of the redundant sections related to generalities of protein structure which does not directly relate to the current topic of research.
  2. Some typos, and merged words to be corrected
  3. Line 85 (and recurrently throughout the manuscript) - FLAG sequence is misspelled, as it should be DYKDDDDK
  4. Line 113 - Why referring only to the removal of toxic target compounds by mNPs?
  5. Line 133 – As mentioned above, unfortunately authors set an incomplete aim, as although the mMIP might be selective for this short FLAG sequence, but its selectivity towards the same sequence bonded to other (longer) recombinant protein sequence(s) is yet unknown, and not at all guaranteed. Therefore, as the main element of novelty and most impactful piece of evidence is that the developed mMIP is really a viable alternative to the anti-FLAG antibody immobilized solid supports for the purification of FLAG-tagged proteins, these experimental proofs are yet to be made, where a clear picture of the offered selectivity may be offered to the readers.
  6. Line 270 – What does it mean “after functionalization with double bonding people”?
  7. Section 2.3.3. - Figure 9 - What is exactly the degree of homogeneity of MNP i.e. assessed by DLS, as in the Figure la lack of a good homogeneity in size is observed? Please comment the differences between mMIPs obtained in organic and aqueous media. Emphasize and justify the choice of organic solvent? How does the use of organic solvent influence imprinting efficiency, selectivity?
  8. Section 2.4 – Justify the selection of functional monomers. Discuss the aimed interaction between the polymer and template (epitope, oligopeptide). In which step the template (oligopeptide) is removed? Peptide loading should have simulated a more relevant scenario, using also FLAG-tagged peptides when assessing MIP selectivity, recovery, etc. It is an oversimplified and idealistic approach to consider only the FLAG-tag itself the targeted analyte, and do not test the by real experiments the experimental hypothesis.
  9. Line 381- as mentioned, specificity is proven on MIPs using different other peptides with variable degrees of complementarity towards the target molecule (FLAG), and not based on the MIP vs. NIP recoveries. How do authors explain the overestimation of recoveries (i.e. 105.6%) in case of DYKDDDDK for mMIP?
  10. Line 387 – In the context of hydrophobic interactions, how would real oligopeptides and polypeptides behave on the imprinted polymers and how would selectivity look like, are still open, unanswered questions.
  11. Lines 394-395: In mMISPE type separation selectivity is even more important in comparison with the recovery values, which unfortunately has not been assessed. Furthermore, no comparisons with the conventional immunoaffinity method has to been shown.
  12. Conclusions – line 606-607 – some conclusions (e.g. the applicability and usefulness of the developed MIP for sensor development) are unsupported by results or discussions.

Reviewer 3 Report

The submitted manuscript reported the core-shell magnetic imprinted polymers for the purification of FLAG-Tagged proteins. The topic is interesting, and the MIPs developed in this work could do a favor to solid phase extraction. I, therefore, think the manuscript can be considered for publication after addressing the issues mentioned below.

1. Introduction needs to be condensed. Some statements are not necessary in this part, for example, characteristics of peptides, unless they are relevant to the main arguments of this work.

2. Molecular imprinting is the core technique of this work. However, its research progress and important applications in wide research fields are insufficient, such as selective separation (10.1016/j.memsci.2022.120750), detection (10.1038/nnano.2010.114), sensing (10.1016/j.cej.2022.138748), and drug delivery (10.3390/polym14051027).

3. Figures 2 - 5 should be integrated together for an intuitive comparison of microtopography, and please provide TEM images with the same magnification. Similarly, it is better to incorporate Figures 9 and 10.

4. Since FT-IR spectra of Fe3O4 and Fe3O4@SiO2 are repeated in Figures 7 and 8, the two figures can be fused together, which will also make it easier to compare the spectra of aqueous MIP and organic MIP.

5. For MIPs, the imprinting factor (IF) is an important parameter to evaluate the performance, which should be calculated and provided (refer to the equation in 10.1016/j.bios.2007.01.017).

6. In general, the adsorption kinetics of MIPs is also important for solid phase extraction, so it should be explored and added in the revised manuscript.

Round 2

Reviewer 1 Report

Dear Editor,

I am happy with the author's comments. The author has added data to the current version of the manuscript.  Also, the author has made significant changes in the manuscript as per the suggestions. I recommend the proof of concept work for the publication.

Author Response

No response is needed

Reviewer 2 Report

The authors have revised their manuscript in many aspects; however, the some critical issues has not been addressed properly. Although authors has brought the title of the manuscript closer to the existing experimental reality (FLAG@mMIP synthesis in both aqueous and non-aqueous media, physico-chemical and instrumental characterization of the mMIP, and selectivity testing with the FLAG-tag, one shorter oligopeptide (DYKD) and two free amino acid), this does not change the final destination of the developed affinity SPE platform, that is using it for the purification of FLAG-tagged polypeptides, which eventually has not been demonstrated experimentally. Since the transferability of MIP synthesis is not a straightforward process, unfortunately one can not rely on the observations of previous studies. The study published in Anal. Chem. 2019, 91, 4100 used as template the tetrapeptide (DYKD) and sacrificial silica beads to enhance porosity of the resulting imprinted polymer, whereas the one published in Applied material and interfaces 2020,12, 49111 used as template the pentapeptide (DYKDC) on porous silica beads.

As throughout the justification of the study the importance of FLAG-tagged peptides/proteins is discussed in comparison with the existing immunoaffinity protocol, the mere separation (purification) of the FLAG-tag oligopeptide (DYKDDDDK) is of limited scientific and no practical relevance, and the purpose of the study has not been attained. Furthermore, comparisons about the reusability of FLAG@mMIP vs conventional anti-FLAG affinity gels would only be relevant if real FLAG-tagged proteins would have been tested (Anal. Chem. 2019, 91, 4100). In the lack of experimental evidence, it may only be hypothesized that in the presence of FLAG-tagged proteins (and other matrix constituents) the reusability of mMIPs would also be severely affected.

Additional comments related to the operated changes:

1. Lines 102-104: not clear why mixing purification procedure with the detection process?

2. English should be improved as often new paragraphs are incomplete, fuzzy or unclear: i.e. lines 143-145; 168-169; 461-462; etc.

3. Lines 172-173: Unfortunately, it has not been experimentally demonstrated that the developed mMIPs do solve all the problems of the conventional SPE procedure (i.e. actual reusability, binding and elution efficiency, etc. when using FLAG-tagged proteins in matrix matched samples, as all these parameters are expected to change).

4. Lines 405-408: In which case the two different interactions ”electrostatic” or “ionic” between the monomer and template molecule are envisaged? What would be the difference between these two types of interactions?  No electrostatic interactions (i.e. ion-pairing) are foreseen in non-aqueous media?

5. Lines 457-463: it is confusing to use selectivity and specificity interchangeably. As mentioned, useful analytical selectivity may not be relevantly quantified based on IF values. What is the recovery of DYKDDDDK in the presence of DYKD? What would be the bias in recovery in the presence of other oligopeptides, FLAG-tagged proteins, etc.

6. Lines 462-463: This is exactly why it is so important to demonstrate the size- and/or functional-selectivity and the real applicability of the mMIPs using a FLAG-tagged polypeptide. As molecular recognition/affinity of MIPs is not a linear feature and it is shaped both by attractive and repulsive forces. Thus, continuing by a simple line of reasoning, a longer polypeptide (FLAG-tagged polypeptide) might not be retained with the expected affinity in the binding cavities due to steric impedance, for example. Moreover, having longer polypeptide chain, non-specific interactions might be more pronounced both on the NIP and MIP. Furthermore, binding and elution kinetics, and therefore biding efficiency (recoveries) are expected to be different when working on FLAG-tagged proteins.

Unfortunately, as already mentioned, only solid scientific data and not hypothetical assumptions are able to experimentally demonstrate the applicability of these mMIPs for their intended use.

7. Citation of references should be checked [17, 19] is [18,20].

 In conclusion, the current reviewer does not recommend the publication of the revised manuscript in the present form.

Reviewer 3 Report

Despite a comprehensive revision made by the authors, there are still some formal problems that need to be improved. In my opinion, the manuscript can be accepted after addressing the following issues.

1. The layout of assembly images (Figure 3, Figure 4, Figure 8) is not acceptable in its current form. Panels need to be arranged neatly and should be adjusted with the same height. Please refer to other published articles.

2. Generally, tags should be placed in the same position in every panel. For those labels with low contrast, changing the font color or adding a solid background is acceptable. In addition, the tag of the second image is missing in Figure 4.

3. There are too many paragraphs in the introduction section, which affect the continuity of the reading. Statements with the same topic should be integrated. The introduction can be divided into the following paragraphs according to the statements of the authors:

i. Introduction of peptides and the necessity for detection

ii. Review of methodology for protein detection and purification

iii. Existing challenges

iv. Feasibility of molecular imprinting technology to breakthrough above challenges

v. Statements of limitations and proposed solution based on magnetic nanoparticles

vi. Achievements in this work

Therefore, Para 3 (The addition of…) and Para 4 (The stability of…), Para 5 (Molecularly imprinted polymers…) - Para 8 (MIPs can be prepared…), as well as Para 9 (Magnetic nanoparticles…) and Para 10 (There are different…) should be integrated.

Round 3

Reviewer 2 Report

It is also not uncommon at all to have divergent views either in-between reviewers or authors-reviewers on given topics or the required complexity of experiments reported, etc.

Based on the provided comments so far and the authors’ response the editor should decide if the manuscript meets the scientific standards of the journal or to invite additional reviewers to express their opinion on the revised manuscript.